# Learning to Infer Run-Time Invariants from Source code

## Abstract

Source code is notably different from natural language in that it is meant to be executed. Experienced developers infer complex "invariants" about run-time state while reading code, which helps them to constrain and predict program behavior. Knowing these invariants can be helpful; yet developers rarely encode these explicitly, so machine-learning methods don't have much aligned data to learn from. We propose an approach that adapts cues within existing if-statements regarding *explicit* run-time expectations to generate aligned datasets of code and *implicit* invariants. We also propose a contrastive loss to inhibit generation of illogical invariants. Our model learns to infer a wide vocabulary of invariants for arbitrary code, which can be used to detect and repair real bugs. This is entirely complementary to established approaches, which either use logical engines that scale poorly, or run-time traces that are expensive to obtain; when present, that data can complement our tool, as we demonstrate in conjunction with Daikon, an existing tool. Our results show that neural models can derive useful representations of run-time behavior directly from source code.

## 1 Introduction

Software maintenance requires reading a lot of code. Experienced developers are adept at this, garnering rich semantics just from this "static" (*viz*, without running the code) inspection to find complex bugs, predict a function's outputs from its inputs, and learn new coding patterns. They strongly rely on generic assumptions about the program's run-time behavior; e.g., that a list index never escapes the list bounds and strictly increases. Such "invariants" capture general, yet relevant constraints on the program's expected run-time behavior.

Automatically inferring invariants can help both developers and tools: first, they can be used to detect bugs where explicit assumptions are incorrect or implicit ones ought to be explicit; second, invariants can guide myriad other tools, such as test-case generators (Artzi et al., 2006). However, inferring invariants is not tractable in general and sound approximations don't scale beyond very small programs. Instead, popular tools either use dynamic trace data from real executions (esp. Daikon (Ernst et al., 2007)), which requires costly instrumentation, or focuses on highly constrained cases such as loops (Sharma et al., 2013a; Padhi et al., 2016).

```
public void onTargetFound(…) {             public void onTargetFound(…) {
  ...                                        ...
  int time = calculateTime();                int time = calculateTime();
  if (time > 0) {                            action.update(-dx, -dy,
    action.update(-dx, -dy,                    time, interpolator);
      time, interpolator);                 }
  }
}
```

*time > 0*

Figure 1: A snippet that demonstrates how explicitly guarded code is often equivalent to code with salient implicit, invariant-like conditions. The code on the right was a real (bug) that was patched by adding the conditional check on the left. We synthesize such samples to train our model by selectively removing if-statements. Our model correctly predicted this repair.

Yet this scalability obstacle may be largely artificial. Practical programs rarely take on an exponential range of values (e.g., integers *tend to* come in a bounded range), and developers seem able to make such inferences without undertaking a project-scale analysis. Rather, they reliably extract them from a local context, using their past experience and cues from the code itself. Consider the snippet in Figure 1: the program on the right uses a `time` variable, returned from one method and passed to another. Not only is 'time' generally non-negative, in this particular case we should not update a position (using moments `dx, dy`) if no time has passed either. This inference, and many more, can quickly be made from reading just these lines of code. Other times, such implicit inferences should be made explicit: this snippet was later repaired by adding the guard on the left.

Based on this observed symmetry between explicitly guarded code and implicit run-time assumptions about code, we propose a model that learns invariants directly from static code. As developers rarely "assert" invariants in their code, we train this model using a proxy, by automatically converting explicitly guarded code to its implicitly guarded counterpart across millions of functions. The generated programs are constrained to be similar to real functions and used to train a large model with a new loss function that is aware of logical constraints.

Our model, BODYGUARD predicts a rich vocabulary of conditions about arbitrary code from new projects, and can be used to find & fix real missing-guard bugs, such as the one in Figure 1, with over 69% (repair) precision at 10% inspection cost. It also predicts more than two-thirds of Daikon's invariants that could previously only be inferred with run-time data, and some entirely new ones that can be validated automatically with trace data. Our work presents a significant next step in learned static analysis, being the first to reliably produce natural invariants from arbitrary code alone. More broadly, we show that learned models can implicitly represent behavioral semantics, just from code.

## 2 OVERVIEW

Inferring invariants for arbitrary programs is NP-hard. Sound approaches using theorem proofers are therefore constrained to restricted settings, such as simple loops (Sharma et al., 2013a), or ones with known inputs (Pham et al., 2017). Such approaches generally don't *scale*: needing SMT solvers limits tools to the few program points where invariants can be proven, and ground-truth inputs typically need to be constructed by hand. An alternative is to use execution traces (Ernst et al., 2007): when realistic workloads are available (e.g. from test suites), they generally span entire systems. However, genuinely representative workloads are rare, so trace-based tools often generate poor invariants (Kim & Petersen). A key concern is that none of these have a notion of relevance, or naturalness of the actual statements (Hellendoorn et al., 2019a).

To address these gaps, we propose a learned invariant generator that predicts directly from code, trained with realistic examples. Our central claim is that the natural distribution of programs includes many groups of similar functions, some of which assert run-time assumptions explicitly, and with much detail, while others vary along these dimensions. As Figure 1 highlights, it is common for code not to state salient conditions (`time > 0`, on the right) that developers may naturally intuit, while other times (e.g. in a later revision, on the left), such conditions are explicitly checked. If this distributional assumption holds in general, then we can use *explicit* conditional checks that guard blocks in functions to teach our models about the *implicit* invariants of unguarded blocks in similar functions. Furthermore, we conjecture that in such comparable samples, the condition is both *salient* (since it is checked explicitly) and *natural* (since it is written by humans). Learning from such examples is thus a very appropriate training signal for inferring practically useful invariants.

Figure 2 illustrates our data generation: we find explicitly guarded blocks in functions that can be removed without substantially perverting the program, and convert these checked cases to implicit ones (Section 3.1). We garner a large aligned dataset to learn to predict the reverse of this mapping, training a Transformer-style model for code, augmented with a loss that encourages sampling logical conditions (Section 3.2). This model, nick-named BODYGUARD, works on any (Java) function, quickly adapting to the local vocabulary and semantics, and has a natural inclination to generate realistic, salient invariants that are often valid (Section 4). This result fits in a long line of observations that programming is remarkably predictable, including in its syntax (Hindle et al., 2012) and execution values (Tsimpourlas et al., 2020), likely by developers' design, to control the complexity of the task (Casalnuovo et al., 2019). Yet none of these relate code and its execution directly, as we do through translating the former into general, intuitively meaningful statements about the latter.

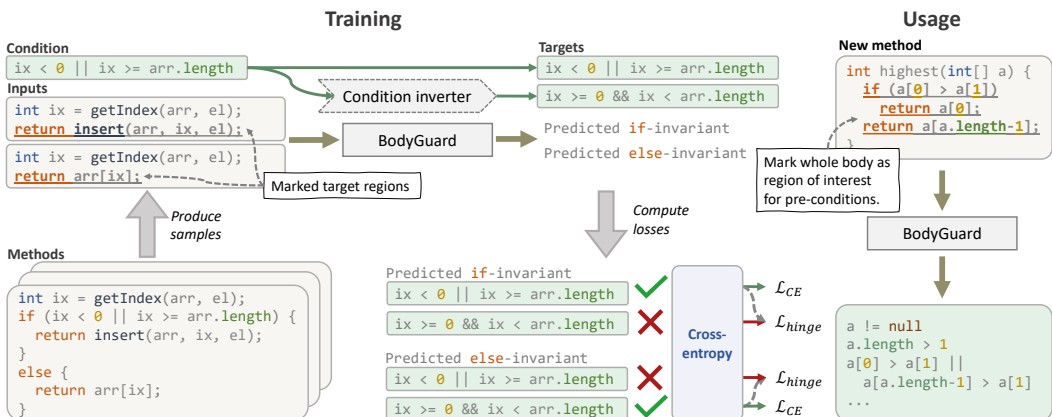

Figure 2: An overview of our learning approach. We extract samples from if statements in Java methods by removing the guard and assigning it (or its negation, for the else block) as the target invariant of the previously-guarded block (if and else blocks separately, if both present) for a translator. We train using the cross-entropy of the predictions given the target, as well as the contrast of this entropy to that of predicting the logical inversion (per sample) in a hinge loss, which encourages BODYGUARD to distinguish between syntactically similar, but logically distinct invariants.

## 3 APPROACH

Training and evaluating this approach required a substantial experimental setup: we collect three datasets for three types of evaluations and introduce an improved loss function. This section describes the data collection, evaluation, and modeling setup generally; Appendices A.1 and A.2 provide additional details on our datasets and modeling architecture, respectively. Our benchmark datasets, code, and models are available at http://omitted.link.

### 3.1 DATASETS

To train BODYGUARD, we generate ca. 2.5 million aligned invariant/function samples from methods with if-statements. We extract these from top-starred Java projects from Github, which we split at the organization level into training (920 projects), held-out (19 projects), and test data (61 projects). Each file was parsed to extract all its methods, from which we generate one sample for each (side-effect free) if- (or if-else-)statement by removing said guard and storing its condition. This produces an equivalent code fragment in which the statement's condition is presumed to either be always **true** (if its body is kept) or **false** (otherwise). Correspondingly, the omitted condition (or its negation) becomes an *invariant* on the remaining code. The resultant sample contains the entire method (minus conditional check) as context, with the *range* of tokens where the invariant condition applies indicated.

We train our model to generate run-time conditions for any indicated segment of code in Java functions. We evaluate its ability to do so in two settings: 1. identifying and repairing missing explicit if-guards, collected from real bug reports, and 2. measuring the validity of our predicted invariants using trace data, collected with Daikon (Ernst et al., 2007). For the first, we collect a dataset of real missing if-condition bugs from across the history of 10K Java projects by parsing all the revisions in these projects' histories and selecting for changes that a) introduce a single if-statement to guard previously un-guarded code, and b) are described as a bug-fixing change (see Appendix A.1.3 for details). We find ca. three thousand of these. For the second evaluation, we use Daikon to collect execution trace data from a smaller set of eight projects that we manually instrumented. We then compare our predictions to both those generated by Daikon, to measure overlap, and to the collected traces directly, to assess the validity of the invariants that we uniquely generate. This helps us understand the *inference gap* between static and dynamic information; i.e., is run-time data (when present) strictly more useful than code, or are the two information sources orthogonal?

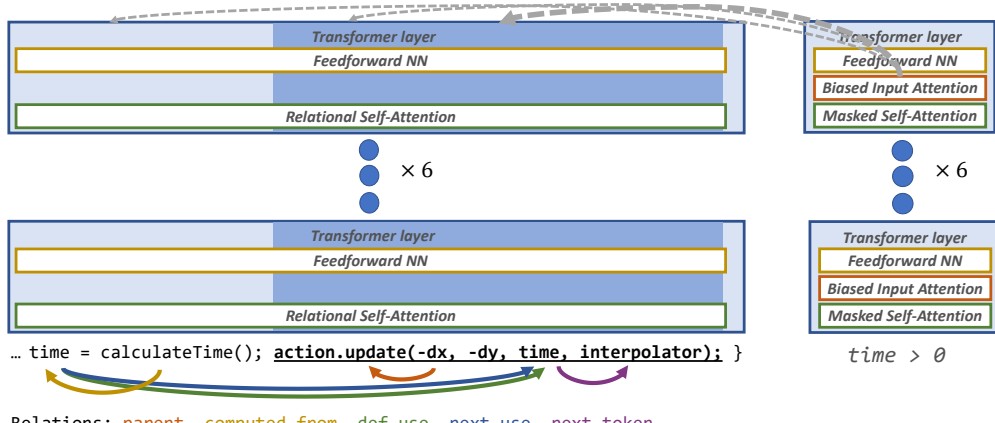

Figure 3: Schematic overview of our model. Both encoder and decoder use 8 Transformer layers. Input is provided as BPE tokens (not shown) augmented with program-graph edge information, which the encoder uses through relational self-attention from (Hellendoorn et al., 2020). The decoder uses both masked self-attention and input attention biased towards the target scope (bold and underlined).

## 3.2 MODEL SETUP

Discovering invariants is non-trivial even for experienced developers, so we both equip our models with substantial capacity and training time, and design to prioritize precision over recall. Figure 3 shows an overview of the architecture, inputs and outputs of our model.

### 3.2.1 ARCHITECTURE

We base our architecture on the Transformer (Vaswani et al., 2017), amplified with the relation attention mechanism from Hellendoorn et al. (2020). While standard (lexical) language models are quite useful for code, Allamanis et al. (2018) and others have shown that utilizing syntatic & semantic information such as the AST, or control/data-flow relations, outperforms text-only models. Hellendoorn et al. (2020) propose a Transformer-based architecture that handles such relations but is faster to train *and* more powerful than graph neural networks (Allamanis et al., 2018). Their model relies on an added attention bias $b_{ij}^r$, injected into the query-key comparison of the Transformer's conventional scaled dot-product attention: $e_{ij} = (\mathbf{q_i} + b_{ij}^r)\mathbf{k_j}^\top / \sqrt{N}$. This bias is sensitive to known relations $r$ between tokens $i$ and $j$ (if any, and summed together if more than one), allowing the model to selectively sharpen (or dampen) the significance of each relation. We adopt this model for our work, specifically with 512-dimensional hidden states, 64-dimensional relational embeddings, 8 attention heads, and 8 layers, totaling ca. 67M parameters.

Our model uses relational information in the form of program graphs. A program graph extractor has been released for C# code (Allamanis et al., 2018), but not yet for Java, so we created our own. Specifically, we extract 5 commonly used edge types, all bi-directional, reflecting common lexical, syntactic, and semantic relations in programs (detailed in Appendix A.2.1). We use the same "leaves-only" representation as Hellendoorn et al. (2020) to limit the size of our inputs by not including non-terminal AST nodes, but instead rerouting edges that connect such nodes to representative syntax token (e.g. from an if-statement node to its "if" token in the code). Finally, to ensure that our decoder is aware of the specified range of code tokens where the invariant applies, we also leverage the relational mechanism between the decoder and encoder, using a simple unary relation (i.e., that a token is part of the invariant's range) between the generated tokens and input tokens.

### 3.2.2 DECODING LOGICAL STATEMENTS

We synthesize training data using a proxy for invariants, which necessarily introduces some bias towards characteristics of if-conditions (and the code they guard) that is incompatible with true invariants. Most notably, in code, small syntactic differences lead to drastic changes in run-time

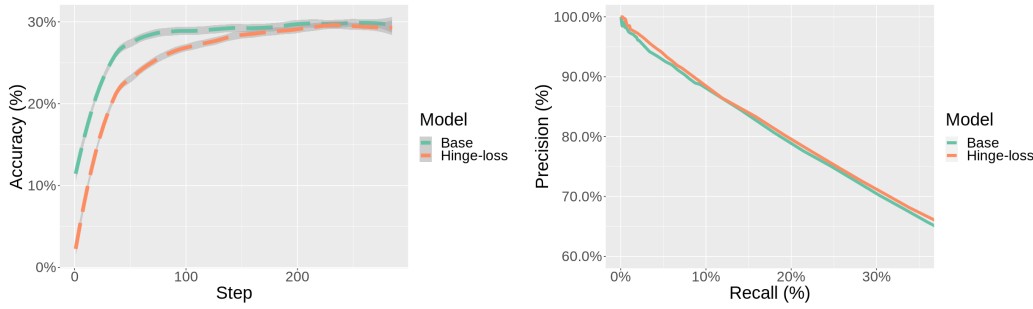

(a) Accuracy on held-out data during training.

(b) Precision-Recall response of the trained models to limits on the entropy of generated invariants.

Figure 4: Model performance during and after training, focusing on the high-precision/low-recall domain for the second (overall test accuracies: 33.8% base, 34.7% hinge-loss).

behavior. It is common for `if`-`else` statements to have quite similar bodies, for which we generate two samples: one with the if-condition as an invariant for the `if` block, and one with its logical negation for the `else` block. This approach tends to produce very similar code fragments with very similar, but logically opposite (e.g. '`!= null`' vs. '`== null`') conditions.

We supervise our model to encourage its representations for *syntactically close* but *semantically opposite* statements to be distinct by introducing a *contrastive hinge loss* term. For every training sample, we produce the logical negation of the invariant and require the decoder to produce that negation with a much higher entropy than the original. Concretely, given a statement $inv$ comprised of tokens $t_i$ and a negating function $neg$, we use the regular cross-entropy loss $\mathcal{L}_{CE}$:

$$\mathcal{L}_{CE}(inv) = - \sum_{i=1}^{|inv|} \log\ prob(t_i \mid t_1 \cdots t_{i-1}, context)$$

to compute the entropy distance w.r.t. its negation:

$$\Delta_{inv} = \mathcal{L}_{CE}(neg(inv)) - \mathcal{L}_{CE}(inv)$$
$$\mathcal{L}_{hinge}(inv) = \max\left(0, \Delta_{inv} - \epsilon\right)^2$$

in which $\epsilon$ is the minimum desired entropy "distance" in bits. In this work, we set $\epsilon = 2$. For this *hinge-loss model*, as we will call it in the rest of this paper, we train with a loss equal to $\mathcal{L}_{seq} + \mathcal{L}_{hinge}$.

## 4 ANALYSIS

We first assess our model's precision/recall behavior on our automatically collected corpus; then, we apply it to a promising down-stream task: missing if-guard repair (and detection), which further helps us assess the models' sensitivity to *salient* invariants. Finally, we use trace data to get a measure of our invariants' *validity* and contrast it with an execution-based tool.

### 4.1 CORPUS DATA

We sample our two models' held-out performance every 100,000 samples while training,[1] leading to the learning curves shown in Figure 4a. The base model saturates earlier than the one employing a contrastive hinge loss, as the latter faces the more challenging task of distinguishing between very similar statements. However, after ca. one week of training, both models converge to approximately the same quality. It speaks to the challenge of the task that the models only reach ∼30% accuracy, due in part to the enormously diverse vocabulary of statements that occurs across our corpus, and to the inherent ambiguity of generating a single invariant when multiple valid options are available (as

---

[1]A full epoch is approximately 2.3M samples for the base model and twice that for the hinge-loss models

Table 1: Bug-detection and repair results on finding and predicting missing if-guards, across two settings: given the correct location, and across all possible locations, further analyzed by aspects of the top prediction.

| Objective | Accuracy | Top-5 Acc. | Precision @10% Recall |
|---|---|---|---|
| *Location given* | 29.3% | 41.9% | 69.1% |
| *All Locations* | 10.4% | 18.8% | 39.9% |
| invariant correct | 15.2% | 24.2% | 48.1% |
| position correct | 19.4% | 43.8% | 100.0% |

we will study later). This task clearly stretches our current models of code to their limits, making it a promising new task to pursue in order to improve our models.

We evaluate each model at the step with their highest held-out accuracy on the test data, where we compare the top generated invariant (from beam search, size = 25) to the ground truth. Figure 4b shows the precision/recall behavior of the two models in the high precision range, which is generally much more useful to developers than high recall. We rank predictions by their entropy: an invariant that is highly likely to be sampled from its context is likely correct. Both models respond strongly to this entropy threshold, becoming especially far more precise when entropy values drop below 1.0 (around 40% recall), and converging to (near) perfect precision, at a commensurate expense of recall. Both break 80% precision at nearly 20% recall, which still accounts for tens of thousands of program points across our test projects alone. Going forward, we use the hinge loss model, which has the better precision-recall trade-off, and prioritize precision over recall.

## 4.2 MISSING IF DETECTION

Using the $\sim$3K real missing if-guard bugs collected from project histories (see Section 3.1), we first measure our model's accuracy and precision at predicting this guard from the localized bug in the top row of Table 1. This most directly related to its training signal, where we provided our model with the location of the code guarded by the targeted invariant. Our model achieves a similar overall accuracy here (ca. 29.3%) as on our general test data.[2], and precision at 10% recall is also quite high (69.1%), allowing us to fix 215 out of 311 bugs at that level once located. That these tasks appear to be comparably "hard" is relevant; automatically synthesized training data is often overly easy compared to real tasks, which harms generalization (Hellendoorn et al., 2019b).

We also care about our model's sensitivity to *salience:* the missing condition in these samples is (arguably) the most important invariant in the entire method, not just the indicated code block. Our model should be able to detect this given how it was trained. This contrasts with tools like Daikon Ernst et al. (2007), which emit all logically valid invariants, many of which irrelevant (Hellendoorn et al., 2019a). The next three rows of Table 1 show the results of running our invariant generator on every contiguous segment (up to 5 blocks) of code in each buggy method, ranking the top invariants across segments for inspection. This is substantially harder than the previous task, reducing the overall accuracy threefold and roughly halving precision. Nevertheless, that is still much better than might be expected if BODYGUARD had no location-sensitivity: we test over 30 blocks per method on average. We also show that the top prediction often matches some aspect of the correct answer, especially the position, and often predicting the correct invariant at another (nearby) block of code.

Finally, we note that the other (low entropy) invariants predicted here are often not at all "incorrect"; from cursory inspection, many are valid, meaningful statements. We study their validity next.

## 4.3 VALIDITY AND OVERLAP WITH DAIKON

Learning invariants just from code stands in sharp contrast to most current approaches in this field, prominently including Daikon (Ernst et al., 2007), which learns invariants from execution trace data instead. Collecting trace data requires instrumenting projects and access to diverse, representative workloads. This makes it much harder to apply to arbitrary code than our approach but has the benefit of offering stronger guarantees of correctness. Comparing our model with Daikon in projects where this information is available thus allows for two useful evaluations. First, we can lower-bound

---

[2]The base model (trained without hinge loss) reached 26.8% accuracy.

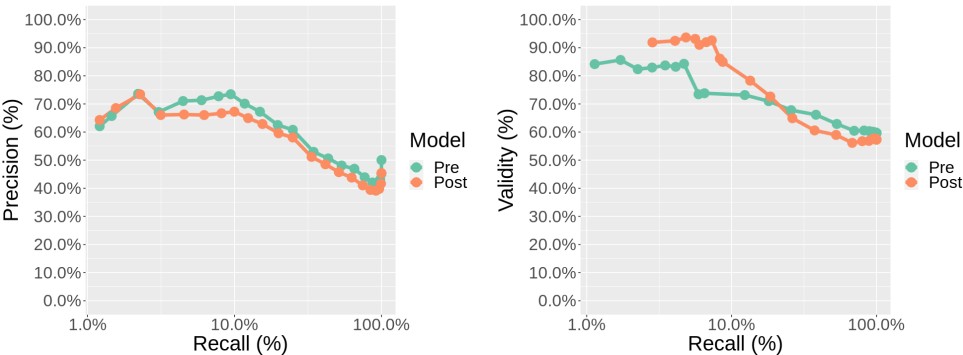

(a) Overlap between our invariants and Daikon's, on pre- and post-conditions.

(b) Overall validity in relation to varying entropy thresholds; pre- and post-conditions.

Figure 5: Results of the overlap and validity analysis of our invariants based on Daikon-extracted trace data. Note the log-scaling on the x-axis.

our tool's true-positive rate by determining how often it replicates Daikon's own invariants, which we tentatively deem "safe" because they hold on all observed traces and have passed a significance test.[3] Second, we can use this trace data directly to determine the validity of (a subset of, see Appendix A.3.2) our invariants that do not overlap with Daikon's.

Figure 5a shows the first result: the frequency with which our invariants overlap with Daikon's, again plotted against recall, where the points correspond to entropy threshold ranging from 1e-4 to 10. Evidently, pre-conditions are easier to predict for our model, likely because it has no real notion of post-conditions (see Appendix A.3.2). Even so, our tool can retrieve more than two-thirds of Daikon's invariants at a respectable 10% recall from static code alone, which is quite promising.

We generate 10 invariants per program point using beam search, so even at a low entropy threshold we produce many pre- and post-condition that Daikon does not (those either out of its vocabulary, or with too few observations). It is reasonable to expect many of these to be valid given previous results. Since Daikon does not provide a means of validating a plain-text invariant, we wrote a simple logical engine that parses Daikon's trace data files and compares a number of categories of our invariants against the recorded values, such as array length, string equivalence, **instanceof** checks, *etc*. Using this approach, we are able to validate ca. 40% (12K) of our emitted invariants, resulting in the validities summarized in Figure 5b. In short, our invariants at full recall are valid ca. 60% of the time, and this validity ratio greatly increases as we sharpen the entropy threshold, to over 80%, at recall values under 10%.

Many of these validated invariants were not produced by Daikon, implying that static and dynamic data are orthogonal for this task. We collected the 708 pre-conditions that BODYGUARD generates at an entropy of $\leq 0.1$; of these, 540 could be checked automatically with trace data, yielding 449 valid and 91 invalid cases. We manually inspected the 168 remaining cases and found that most (122) were valid, but Daikon's tracer simply did not record the information needed to predict these.[4] Overall, this suggests that more than 80% of our invariants at this recall level (3.5%) are correct, and more than two-thirds of the invalid remainder could be ruled out using trace data, if available, leaving a false positive rate of just 6.5% (46/708) when execution data is available (while also adding about 200 valid invariants to Daikon's own predictions). This supports our belief that our tool is largely orthogonal to, and usefully synergistic with, dynamic, trace-based invariant generators.

---

[3]Though in practice it generates a fair number of spurious statements still.

[4]Some of these were correct statements but not proper pre-conditions, e.g. invariants about a variable declared at the first line of the function. This is an artifact of our training setup, which has no explicit notion of method-level pre-conditions. We marked these as invalid for this analysis.

## 5 RELATED WORK

Automatically inferring invariants is usually approached either in constrained settings where some "checker" (e.g. an SMT solver) or ground-truth is available, or under the assumption that we have access to execution traces from realistic workloads. Among the first, Sharma et al. (2013b) find algebraic (polynomial) invariants by solving a system of linear equations with an SMT solver and using counterexamples to create new test inputs. Sharma et al. (2013a) use PAC-learning to learn integer loop invariants on programs with a single loop, trained by contrasting passing and failing test cases. Padhi et al. (2016) learn pre-conditions and loop invariants as boolean combinations of arithmetic conditions ("features"), which they synthesize by generating and testing all features up to a size cutoff. This approach is agnostic to the program structure, as is Pham et al. (2017), who use a fixed set of feature templates over state vectors to learn linear inequalities that classify passing and failing state vectors, requiring both post-conditions and passing and failing tests to be in place. In contrast, our work makes no assumptions about the code other than the availability of a parser. In settings where an SMT solver (or test cases) is available, it could be used to filter invalid invariants generated by BODYGUARD.

Among machine learning based approaches, Si et al. (2018) use policy-learning to teach a GNN to generate loop invariants in cooperation with an SMT solver (Z3), which provides intermediate rewards (through counterexamples) to finesse the sparsity of the eventual reward (the final validity of the invariant). A second reward is added to reject "meaningless" and "trivial" predicates such $e$ $== e$ or $e < e$. Besides not requiring an SMT solver, our approach learns notions like "relevant" and "natural" directly from real code. Relatedly, Brockschmidt et al. (2017) also use GNNs to induce invariants over data structures, using a similar approach of generating invariants (in separation logic) supervised by data produced from test runs. The production is based on hand-engineered features over the data-structure graphs. Both these approaches may be symbiotic with ours where tests or logical constraints are known, although they consider different classes of invariants.

Daikon (Ernst et al., 2007) belongs to the second class of invariant predictors, leveraging execution traces from realistic inputs to infer a large vocabulary of method pre- and post-conditions. This general applicability has led to its frequent as a basis for other tools, often to generate an initial corpus of invariants for tasks such as automated patching (Perkins et al., 2009) and test generation (Artzi et al., 2006; Pacheco & Ernst, 2005). However, truly representative inputs are rare, and using incomplete data risks generating many irrelevant or invalid invariants. Polikarpova et al. (2009) found that the size of the test suite affects the validity of generated invariants on Eiffel programs. Kim & Petersen anecdotally note various issues with Daikon's invariants on large, C++ systems, such as a high degree of false positives and few insightful invariants. Hellendoorn et al. (2019a) similarly observe (on hand-annotated C# functions) few relevant and valid invariants based on executions from unit test. Our approach learns directly from natural conditions to generate relevant and generalizable conditions, and when trace data is present, it can be used to filter out invalid invariants.

## 6 CONCLUSION

We conjectured that typically used invariants are in a sense *natural*, like many other aspects of programs (Hindle et al., 2012; Barr et al., 2013; Tsimpourlas et al., 2020), and therefore predictable, intentionally written in standardized ways for ease of reading and writing Casalnuovo et al. (2019). Our results support this claim: both explicit (if-statements) and implicit (invariants) conditions pertaining to code can be predicted precisely, and with high validity from code reading alone, facilitated by our proposed data generation approach and loss function. As a result, we can generate many invariants that were previously only accessible through trace data (and more), which greatly increases the reach and applicability of invariant inference.

This finding has broad implications: our tool can provide valuable semantic insights both to developers, e.g. to aide debugging efforts or facilitate code understanding, and to other tools, many of which struggle to navigate an exponentially large search space of programs. Our tool can help bias this search space using highly likely assertions, which could greatly improve the range and quality of solutions found by downstream applications. In summary, our novel approach learns to reason about program state by synthesizing training data from if-conditions; this empowers BODYGUARD to reliably generate useful invariants entirely from static code.

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

# A  APPENDIX

## A.1  DATA COLLECTION DETAILS

We base our evaluation on a Java dataset consisting of the top 10,000 most-starred Java projects on Github, collected March 30th, 2020 using the Github v3 API. Since generating our training data samples is quite expensive, we used just the top 1,000 (most starred) of these projects to automatically generate training and evaluation samples for the results described in Section 4.1. This dataset was split between training, held-out and evaluation sets at the organization level to ensure minimal duplication, as projects within the same organization often share many coding patterns (Allamanis, 2019). We allocated 95% of organizations (920 projects) to training data, 2% to held-out data (19 projects), and 3% to test data (61 projects), to assess the final trained models.

### A.1.1 INVARIANT GENERATION

We parse each file using Eclipse's JDT parser and extract all (non-nested) methods from the resulting parse tree. Within each method, we detect all if-statements, removing all those whose conditions contain side-effects (such as assignments, increment/decrement operators, and non-whitelisted methods, see Appendix A.1.2), and those whose body contains a control-flow altering statement (e.g. **return**, **throw**) unless it is the sole statement.[5] For the remainder, we generate samples based on the following types of if-statements:

**Simple if-statements:**   these include samples like Figure 1, in which a single if-statement guards a simple body with no control-flow altering code.

**If-else statements:**   for these we generate two samples: one in which we remove the **else** block entirely and generate an if-invariant as above, and one in which we negate the condition and generate an invariant for just the **else** block. Note that **else if** statements in Java are treated as nested statements and thus handled the same way.

**Control-flow altering if-statements:**   any if-statement whose body prevents the execution of subsequent code, by containing just a **return**, **break**, **continue**, or **throw** (Exception) statement, is treated as declaring an invariant (namely, the negation of the if-condition) for the ensuing code.

In all cases, the surrounding context is the entire method, and the *range* of tokens to which the condition applies (namely, those that used to be guarded) is stored with the sample. We generate samples for all these conditions, producing a new sample for every if-statement. This ensures that each sample minimally alters the original code, which reduces the risk that we produce unnatural code (which would harm the generalization of our model). As such, a method can produce many samples, so functions with many conditions will be represented proportionally more often. We do not consider that problematic, as 1. long functions tend to have correspondingly more invariants, so the increased emphasis should be beneficial to our model, and 2. we anyways cap our training samples to only modestly large functions (up to 500 sub-tokens, which typically translates to the order of 20 lines), due to memory constraints.

### A.1.2 PRODUCING NATURAL FUNCTIONS

Not all if-guards can be removed without changing the semantics of the code; conditions can have side-effects. This includes assignments (e.g. **if** ((x = y) != **null**)), certain operators (*viz.* ++ and --) and method calls with side-effects. To ensure that the converted code is semantically coherent, and because invariants should not have side effects anyways, we omit all such cases. Many method calls do not have side effects, so to avoid limiting our dataset too much, we heuristically select a large, but relatively "safe" set of these based on common coding patterns. This includes common "getter" methods, java.lang.Math calls, object equality tests, collection inspection methods, such as inclusion checks (e.g. 'contains', 'has') and size-related methods, and a few miscellaneous others that were common in our training data (e.g. parseInt, name). The regexes used to detect these various types of methods are listed in Table 2.

Removing if-statements does not always yield meaningful code, consider:

```java
int foo(int x, int y) {
  if (x > y) {
    return x;
  }
  return y;
}
```

---

[5]When an if-statement body terminates the current branch of execution only after first executing some other code, generating equivalent unguarded code is complicated: inlining the guarded code (minus the final statement) would often produce very unnatural code, as it tends to involve some form of error-recovery, such as logging or resetting a value. Omitting the entire block instead, as we do for simple control-flow altering statements may be more appropriate; future work can explore this, and various other, corner cases to generate more samples.

Table 2: Categories of, and Regexes used to detect, whitelisted method calls, which are generally assumed not to alter program state and thus allowed in invariants.

| Category | Regexes (space separated) |
|---:|---|
| Object comparisons | `equals([A-Z].*)? deepEquals match(es)?`
`compare([A-Z].*)? hashCode` |
| Collection methods | `length size capacity depth keySet indexOf` |
| "Getter" methods | `(get|contains|has|is|exists|should|can|was)([A-Z].*)?`
`.+Exists` |
| String methods | `(starts|ends)With contentEquals toString substring`
`toCharArray valueOf charAt to(Upper|Lower)Case` |
| Numerical properties | `(boolean|int|float|double|long|byte|short)Value`
`parse(Boolean|Int|Float|Double|Long|Byte|Short)` |
| Mathematical calls | `log([0-9]+)? abs sqrt pow max min sin cos tan round`
`ceil floor` |
| Misc. others | `peek group (.+N|n)ame child.*` |

If we remove the conditional check, the resulting method is left with just two consecutive **return** statements, which is invalid in Java. This particular case would trigger a compiler error, but not all inappropriate removals do: if the if-body had instead assigned `y = x + 1;`, removing it would result in `y` always being assigned `x + 1` before returning, making the parameter useless. Not using a parameter is not erroneous by definition, since the method `foo` may be inherited (or overriden in a subclass) and other instantiations do make use of it, so Eclipse's parser just emits a warning. Since both these cases result in code that is both unrepresentative of typical Java, and would yield highly predictable invariants, we additionally reparse each resulting function after removal of the targeted if-statement and discard any changes that trigger compilation warnings and errors.

Specifically, Eclipse JDT requires full type resolution to guarantee correct program analysis and stops checking for violations if it finds compile-time errors from missing types. When processing as many projects as we do (many of which cannot be built automatically), we cannot soundly resolve all dependencies for each project. As a close approximation, we instead parse each function in its entire project context to allow as much heuristic type resolution as possible. Then, we look for any increase in warnings and errors between the method before and after removing an if-statement. This reduces the number of collected samples and increases the time to generate the dataset (to ca. 200 CPU hours for 1K projects), but also increases its validity by eliminating many inappropriate fragments.

Finally, we limit our functions to those having 500 (sub-)tokens or less to facilitate a reasonable modeling throughput. This does not reduce the dataset by much; most functions tend to fit this limit. In total, we collect ca. 2.34M training samples, 12.1K held-out samples and 101K test samples, with approximately 200 sub-tokens per function on average.

### A.1.3 COLLECTING "MISSING IF" BUGS

We collect our dataset of missing if-condition bugs from across the history of all the aforementioned 10K projects in our dataset. For each project, we parsed every commit to the main branch, using git's "diff" function to identify cases in which the sole addition was to wrap one or more existing statements in an if-statement. This yielded 32,471 samples from across 8,174,552 commits. Although all of these may constitute interesting samples, we prioritize bug-detection for now as the most direct application of our model. To ensure that our collected samples are likely bug-related, we focus only on the ca. 3.7K cases in which the entire commit introduced just a single if-statement in a single Java file and the corresponding commit message contained any of the common bug-related terms such as "fix", "bug", and "fault" (Ray et al., 2016). We additionally filtered out any commits to projects that were included in our training dataset to avoid the risk of overlap (which need not be present as many commits reflect now out-dated code), yielding 3,146 samples in total.

Table 3: Statistics of Daikon's invariants collected on Dacapo projects.

| Project | Methods | Invariants |
|---------|---------|------------|
| Batik | 6,797 | 64,298 |
| Eclipse | 2,375 | 41,257 |
| H2 | 2,169 | 28,965 |
| lusearch | 1,594 | 81,291 |
| luindex | 1,004 | 9,822 |
| PMD | 5,026 | 274,638 |
| Tomcat | 3,983 | 63,146 |
| *Total* | 22,948 | 563,417 |

### A.1.4 RUNNING DAIKON

Comparing our tool to Daikon (Ernst et al., 2007) required some adaptations. Daikon requires projects that are fully built, instrumentable, and have representative workloads. Unit tests are often insufficient because they test for both appropriate and inappropriate values (e.g. those triggering an exception), which is counter to our purpose.[6] Scaling Daikon to our aforementioned dataset is not feasible; indeed, to the best of our knowledge there is no large public dataset of Daikon invariants on real programs. Instead, we created a modestly large dataset of our own.

To do so, we leveraged the Dacapo benchmark (Blackburn et al., 2006). Originally created to benchmark program optimizations (e.g. through better compilers), each project in this benchmark comes with a set of representative workloads designed to execute many of its paths. This is ideal for our case. Practically, although the benchmark comes with a single runner for each project, Daikon could not instrument through the reflective calls that this framework uses. Instead, we manually instrumented and ran 8 projects (details in Table 3) in this suite directly, which, in nearly all cases, involved writing our own "runner" to mimic Dacapo's instrumentation while calling the requisite project-code directly. We then applied Daikon as usual, running the code under instrumentation first and then producing invariants from the resulting traces. Table 3 summarizes the resulting invariant counts.

We limited the volume of the collected trace data by exponentially decreasing the number of traces for each program point once it was seen sufficiently often (10 times) and excluding many values from tracing, such as those that are not visible from the program point of interest and any nested values with more than three levels. Even then, Daikon required upwards of 30GB of RAM and nearly an hour of processing for the larger projects – much more than our models.

## A.2 MODELING DETAILS

### A.2.1 PROGRAM GRAPH EXTRACTION FOR JAVA

We used Eclipse's JDT parser with approximate name-binding resolution to extract five edge types across 3 broad categories of information that are accessible in source code:

• **Lexical:** every token is connected to its neighbors through *next-token* edges (and their reverse). This adds additional sensitivity to lexically local information beyond the positional encoding used in the standard Transformer.

• **Syntactic:** we extract all AST parent-child relations, which provide insight into the hierarchical structure of source code.

• **Data-flow:** we include three types of data-flow edges: *next-use* edges, which connect lexically sequential uses of the same variable; *computed-from* edges, which connect any variable usage to the last value it was assigned, and *def-use* edges, which connect every variable usage to its (single) original declaration point.

In addition, every edge type has a symmetric, mirrored version (e.g. prev-token), yielding a total of 10 distinct edge kinds used by our model.

---

[6]In addition, Daikon cannot instrument JUnit-tested code since it uses reflection, which effectively makes Java tests off-limit.

### A.2.2 TRAINING DETAILS

Consistent with recent observations regarding effective modeling of source code vocabulary (Hellendoorn & Devanbu, 2017; Karampatsis et al., 2020), we use Byte-Pair Encoding to create a sub-token vocabulary based on the tokens in our training data. Our vocabulary, estimated from the training data, spans 10,000 sub-tokens; both the input function and the predicted invariant are sub-tokenized using this (reversible) dictionary. Transformer models generally scale in memory needs with the square of the size of their inputs. To ensure that our minibatches are sufficiently large to keep the gradients stable, we restrict our inputs to functions with up to 500 (BPE) tokens and our invariants to 50 tokens (although invariants that long are very rare). With these cut-offs, we train batches of up to 12,500 tokens in parallel across two NVidia RTX Titan GPU's with 24GB of VRAM each. By packing similarly sized functions per batch, we minimize the overhead from padding and are able to fit ca. 70 functions per batch on average.

### A.3 EVALUATION DETAILS

### A.3.1 IF-CONDITION LOCALIZATION & REPAIR METRICS

Since some methods have far more program blocks than others, simply ranking all invariants across method boundaries by entropy would lead to bigger methods being highly disproportionally represented. Rather, we try to balance method and invariant level *inspection cost* by simulating the inspection of 10% of invariants in our dataset from a subset of methods. We do so by first ranking methods by the entropy of their top invariant, from low to high, and then inspecting all invariants from these methods in order until we have inspected 10% of all location/invariant pairs in this dataset (which number 73,738). The 10% inspection (recall) level in Table 1 correspond to a threshold of just 0.0233 bits, under which the average method has 55.3 blocks – substantially more than the average method overall. Separating out the functions with 32 or fewer program points (the mean), the overall accuracy increases to 16.3% and the 10% recall precision increases to 50.0% – the joint task is naturally easier on shorter methods.

### A.3.2 GENERATING PRE- AND POST-CONDITIONS WITH BODYGUARD

The comparison with Daikon invariants comes with an important caveat: Daikon only generates method pre- and post-conditions. This means that we cannot perfectly classify the validity of all our invariants. Nevertheless, our experiments on missing conditions show that our models are precise at inferring even very specific missing conditions, which strongly suggests (as our manual analysis has too) that many of its other suggestions are valid as well.

Secondly, our tool produces invariants for any syntactic block of code throughout the method and does not have a general mechanism to indicate that pre- or post-conditions are required. To imitate these for our tool, the closest approximation is to mark the entire method body as needing an invariant when a pre-condition is required and the final (`return`) statement otherwise. To avoid the complexity of having to match multiple return points, or none at all for `void` methods, we restrict the latter case to methods with a single `return` statement only. Note that the latter is an imperfect approximation: our tool only learns to predict guards that precede a statement. A guard that it predicts for a `return` statement may not be an appropriate substitution for true post-conditions but rather a reason to return at that particular point.

### A.3.3 MEASURING OVERLAP WITH DAIKON'S INVARIANTS

We quantify the overlap between our predicted invariants and Daikon's using normalized Cumulative Gain. This metric captures the quality of a ranker in terms of how often it returns relevant elements; it is traditionally used in information retrieval, for example to evaluate a web searcher. Although *discounted* cumulative gain is more commonly used, we refrain from penalizing based on "rank" of predictions, because there is no reason to assume that Daikon's invariants are more salient or relevant than others that we predict. That is, all that matters is that Daikon's invariants are *among* our (top 10) predictions.

A.4 FURTHER RESULTS

A.4.1 CHARACTERISTICS OF MANUALLY INSPECTED INVARIANTS

A large portion of the manually verified invariants in Section 4.3 corresponded to fairly trivial statements, such as **instanceof** assertions for a value being cast to the corresponding type. In some cases, our invariants were more general or accurate than Daikon's; e.g. when BODYGUARD asserts that an object is not **null** whereas Daikon asserts that a member of that object is not **null**. At other times, we inferred invariants that Daikon missed entirely, likely due to limitations in its internal rules and heuristics. For instance, as a pre-condition of:

```
static ReliableFile getReliableFile(File file) throws IOException {
  if (file.isDirectory()) {
    throw new FileNotFoundException("");
  }
  return new ReliableFile(file);
}
```

BODYGUARD correctly inferred that `!file.isDirectory()`, while Daikon only offered `file != null`.

In another case, our tool produced a more specific invariant for this PMD snippet:

```
public int getPriority() {
  return priority;
}
```

Here, Daikon asserts that `priority` level is exactly either 2 or 3, because those are the only observed values in the (evidently unrepresentative) traces off this method. This indicates how Daikon's invariants can be inaccurate even with available workloads. BODYGUARD more broadly anticipates that `priority >= 0`, which matches the method's actual specification as encoded in its Javadoc documentation (which our tool does not use).

A.4.2 FURTHER EXAMPLES

In the below example,[7] a `badge` variable, initialized to **null**, is first assigned a value based on program state, and then added to two collections (`local` and, conditionally, `global`). This second segment, after the **switch** statement, should have been guarded by a check that `badge != null`, since not every **case** assigns it a value. Across all 53 permutations of code blocks (and countless options per block) in this method, BODYGUARD predicts this condition at the correct location at rank 3. Its first prediction was the nonsensical statement `!global` as a guard for the entire method body. Possibly, no good prediction was possible for that range, so this option had low entropy by sheer contrast with other possibilities. The second ranked prediction was `badge == null` for every line after the declaration of `badge`. While this is tautologically valid as a pre-condition for those lines, it highlights the importance of specificity in range – it is only truly *invariant* for some of these lines, specifically, the start of each **case** and the **break** statement of the latter two, a range that is not currently supported by our approach.

```
public static void validateTutorial() {
    Badge badge = null;
    switch (Dungeon.hero.heroClass) {
        case WARRIOR:
            badge = Badge.TUTORIAL_WARRIOR;
            break;
        case MAGE:
            badge = Badge.TUTORIAL_MAGE;
            break;
        case ROGUE:
            break;
        case HUNTRESS:
            break;
```

---

[7]Repaired in https://github.com/00-Evan/shattered-pixel-dungeon/commit/475d78cd0599a1d39c4708a91fbb30c95b3f3418

```
    }
    local.add(badge);
    if (!global.contains(badge)) {
        global.add(badge);
        saveNeeded = true;
    }
}
```

The following snippet[8] returns a default image, generating it on the first call. Even though the documentation of `createBitMap(int, int, Bitmap.Config)`[9] does not specify it, this method can return **null** in rare circumstances, such as when a phone runs out of memory and recovers by aborting this call.[10] BODYGUARD correctly infers `empty != null` as the top invariant, having seen similar calls in other Android projects in its training data. Specifically, it predicts this invariant both for the just the line containing `empty.eraseColor` (rank 1), and for the block including that and the next line (rank 2). The latter is the more correct segment.

```
private static Bitmap getDefaultThumbnail() {
    if (defaultImage == null) {
        Bitmap empty = Bitmap.createBitmap(160, 200,
                              Bitmap.Config.ARGB_8888);
        empty.eraseColor(Color.WHITE);
        defaultImage = paint(empty);
    }
    return defaultImage;
}
```

## A.5 LIMITATIONS

We evaluated our predictions broadly to assess both their salience and validity. Even so, it is hard to automatically assess all of our invariants, especially those inserted in the middle of methods and those whose vocabulary is outside of what Daikon finds. However, the results on the task of predicting missing if-statements, (which avoids these evaluation problems) are quite encouraging; we believe that this bodes well for the more general settings. Future work may better assess validity of our entire vocabulary of invariants, perhaps by injecting asserts corresponding to our predictions into the source code and executing the tests.

Our second main criterion is salience: our predictions should be particularly relevant to the referenced code, in contrast to prior work. We chose to assess this by using real missing if guards, which would appear to be a good example of particularly salient implicit conditions (as developers chose to make them explicit). We did not quantitatively study other types of salience, such as which conditions are most informative or intuitively obvious to real developers. This, too, may be a fruitful area for future work; human subject studies involving invariants have produced worthwhile insights into developer behavior in the past (Staats et al., 2012).

---

[8]Repaired in `https://github.com/SufficientlySecure/document-viewer/commit/680650556340aa15502e1ec375e4255c1c16fb5b`

[9]`https://developer.android.com/reference/android/graphics/Bitmap#createBitmap(int,int,android.graphics.Bitmap.Config)`

[10]As suggested at `https://stackoverflow.com/a/14778533`.

