# OpenReview forum: "Learning to Infer Run-Time Invariants from Source code"
_ICLR.cc/2021/Conference — Reject_

### Official Review · AnonReviewer4 · 2020-10-27
**interesting idea, execution leaves a lot to be desired**

**Rating:** 5
**Confidence:** 5

**Review:**

### Summary ###
The paper presents a technique for inference of certain kinds of program invariants directly from the program’s source code. The basic idea is to treat conditional statements as hints for facts about the program state that should hold at a given program point.


### Strengths ###

* This is a challenging problem and the paper shows some successful examples. The idea that some useful invariants can be inferred based on local information, while not new, is interesting and can lead to follow up work of practical value.

* The contrastive hinge loss of syntactically close but semantically opposite statements is interesting.

### Weaknesses ###

* The paper falls short on the framing of the invariant inference problem, and on the technical details of what does it mean to infer a meaningful local invariant. Starting from trivialities like the fact that the problem is generally undecidable (and not as stated in Section 2), through the use of incorrect terminology for invariants, guards, pre/post conditions, etc. This just makes the paper hard to follow.

* Fundamentally, beyond simple invariants (array bounds, nullness checks) it is not clear why program invariants would generalize well across different programs. The exception is of course the use of libraries and invariants in library contracts (as learned in [PLDI19a, PLDI19b]). For nullness guards, you should take a look at [https://arxiv.org/pdf/1902.06111.pdf]. I think it would improve the paper if you could focus on a certain kind of invariants, and show that these invariants can in fact generalize across programs.

* As a concrete example, take your own Figure 1. Assuming that these are two different programs, there is no reason to assume that the contract of `calculateTime()` remains the same. Had `calculateTime()` been part of some standard library shared between programs, the case for generalization would have been much stronger.

* There has been so much work on static inference of invariants that it is impossible to list even all the closely related work. Some things that are worth looking into are the work on Scalable static analysis [Scaling], the inference of necessary preconditions [Logozzo], and bug detection that is based on "belief" [deviant, belief], which is closely related to your intuition about naturalness and human-written invariants. Also helpful to look at [loopInvariant] and the related work mentioned there.

* Comparison to Daikon. As you correctly point out, Daikon infers likely pre/postconditions. The description of how you compare your invariants to those inferred by Daikon is not clear unless all relevant cases related to (pre)conditions on method parameters.


### Questions for Authors ###

* It would be helpful to see more characteristics of the real missing if conditions that you have collected. I am wondering if these are simple conditions of the kind of missing nullness checks or missing array-bound checks. The way in which you have collected these samples is likely to create a bias towards simple missing conditions. How many terms are in these conditions? How many of them are nullness checks? How many are array-bound checks? How many include simple string operations and/or other simple method calls as implied by Table 2?


### Improving the Paper ###

1. I liked the idea of removing conditionals to infer likely necessary preconditions. It would help to clarify when what you predict is a guard, a precondition, an invariant, or something else.

2. You are clearly not trying to infer any loop invariants, and it would help clarify that upfront.


### References ###

[PLDI19a] Scalable taint specification inference with big code
https://dl.acm.org/doi/10.1145/3314221.3314648

[PLDI19b] Unsupervised learning of API aliasing specifications
https://dl.acm.org/doi/10.1145/3314221.3314640

[Scaling] Scaling static analyses at Facebook
https://dl.acm.org/doi/10.1145/3338112

[Logozzo] Automatic inference of necessary preconditions
https://link.springer.com/chapter/10.1007/978-3-642-35873-9_10

[deviant] Bugs as deviant behavior: a general approach to inferring errors in systems code
https://dl.acm.org/doi/10.1145/502034.502041

[belief] Static error detection using semantic inconsistency inference
https://dl.acm.org/doi/abs/10.1145/1250734.1250784

[loopInvariants] Learning Loop Invariants for Program Verification
http://papers.nips.cc/paper/8001-learning-loop-invariants-forprogram-verification

---

### Official Review · AnonReviewer2 · 2020-10-28
**Learning to Infer Run-Time Invariants from Source code**

**Rating:** 5
**Confidence:** 3

**Review:**

Summary: This paper proposes a novel approach for training a Transformer model to predict program invariant. The model is trained using training data that are synthesized from explicit conditional checks in functions and is used to predict invariants of unguarded blocks in similar functions.

Strength
1. The paper addresses the important and challenging problem of program invariant generation from static code in a scalable way.
2. Real-world “missing if-guard” bugs are detected using the proposed model.

Weakness
1. The idea of synthesizing training data by automatically converting explicitly guarded code to its implicitly guarded counterpart is interesting. However, the effectiveness of the trained model to infer program invariants in a general way is not clear from the experimental results. The evaluation with real-world bugs focuses on “missing if-guard” bugs. The difficulty of detecting this bug cannot be understood as there is no accuracy results from an existing tool (for example, Daikon) in detecting this real-world
2. Although a comparative analysis with Daikon is presented, the presented approach focuses on a narrower class of invariants compared to Daikon. Moreover, Daikon relies on execution traces. A comparison with an existing ML based approach using static code (e.g., [1]) would provide more interesting insights about the model’s accuracy.
3. A contrastive hinge loss is introduced to address the “syntactically close” but “semantically opposite” cases. However, from Figure-4, it seems the performance of the model is not impacted in a significant way by the loss function.

[1] P. Garg, D. Neider, P. Madhusudan, and D. Roth. Learning Invariants using Decision Trees and Implication Counterexamples.

Question to author:
Please address and clarify the cons above.

---

### Official Review · AnonReviewer1 · 2020-10-28
**A new application of machine learning for invariant generation**

**Rating:** 5
**Confidence:** 4

**Review:**

The paper proposes to discover likely invariants for code by observing snippets that check for the given conditions and assuming these conditions encode invariants for the code executing before and after the condition check was checked to hold (respectively not hold for negated invariant). This is a novel idea that uses code with correct if conditions to guess the invariants for code that has the conditions missing.

My main criticism for the paper is that it does not give a compelling reason why one would want to apply this technique. While this is a smart way to obtain the invariants, the paper does not give too much intuition why it could be useful in practice. Even on the examples in the paper, the machine learning algorithm probably learns invariants from identifier names and not from the semantics of  the code around.

The authors can relate the work to a large corpus of learning invariants for functions based on things like usages of functions, e.g. like done in [1] or [2] and the techniques there find actual bugs in code. For example, if the invariant is non-nullness of x, this may be because x comes from a function that sometimes returns null or because it comes from a function that does not accept null. If I would want to do for example bugfinding, I would want to know contradicting invariants coming from the two functions.

In terms of execution, the paper is well written and the techniques look state-of-the-art from a machine learning perspective (although there are no baselines given). However, the experiments are insufficient for showing usefulness of the idea. With Daikon overlap in the 70% range and precision also in the same range, it is not clear that the tool gives any new valid invariants on top of Daikon. In terms of bugfinding, the results are also inconclusive that any bugs can be found. If I would put the tool to test 100 methods, where normally less than 10 of them are buggy, I can expect 20 false positives.



Minor:
theorem proofers -> theorem provers
Figure 5 a talks about overlap, but axis says precision.

[1] Ted Kremenek, Paul Twohey, Godmar Back, Andrew Y. Ng, Dawson R. Engler:
From Uncertainty to Belief: Inferring the Specification Within. OSDI 2006
[2] Insu Yun, Changwoo Min, Xujie Si, Yeongjin Jang, Taesoo Kim, Mayur Naik: APISan: Sanitizing API Usages through Semantic Cross-Checking

---

### Official Review · AnonReviewer3 · 2020-11-03
**Ok paper but lacks evaluation**

**Rating:** 3
**Confidence:** 4

**Review:**

The paper presents a method for statically learning code invariants from source code using a variant of transformers.

Strengths
----
- The paper demonstrates that on the synthetic dataset the proposed approach can infer many invariants.

Weaknesses
----
- The evaluation with a synthetic dataset seems very weak. “If” checks are not good proxies for useful invariants as in most programs there are many if checks that are simply unreachable or redundant. In practice,  not all invariants are useful for the downstream tasks (code fixing, bug finding, etc.) mentioned by the authors. Without a more direct evaluation, it is very hard to tell how useful the learned invariants actually are for these tasks. The paper will be significantly stronger if the authors can evaluate their tool against existing loop invariant inference datasets with ground truth data like those used in Si et al. (NeuRIPS 2018).

- The transformer-based model seems to be directly reused from Hallendoorn et al. (ICLR 2020). Thus the contribution in terms of model design is limited.

- The authors also did not cite/compare against the current state-of-the-art loop invariant learning work: CLN2INV: Learning Loop Invariants with Continuous Logic Networks. Ryan et al. ICLR 2020

---

### Author Response · Authors · 2020-11-24
**General Response**

We thank the reviewers for their feedback. At this point, we are not asking you to change your scores, but would like to respond to general concerns and clarify some comments here. We will take into account the feedback received while revising this work.

First, we underscore that the goal of our approach is quite distinct from virtually all prior work in that it aims to reason about program state at _arbitrary locations_. Several of the responses focused on comparisons to loop invariants, library contracts, and pre/post-conditions. We agree that these are all useful and significant applications, and indeed, technically, our model could be used to predict all of these (and we compared with Daikon in great detail to prove as much). However, it is not our goal to surpass established work in those, comparatively restricted domains, because the unique characteristics of those tasks mean that models designed specifically for them are naturally more useful there. For instance, in the loop invariant case, there is often the possibility of proving the validity of a prediction, which means that models that are designed for those settings can rely on e.g. SMT solvers and iterative/guided generation of invariants. This is clearly a useful (and well-covered) task, and we do not expect to beat established models that can rely on guided search. Yet, by symmetry, those models are not at all as broadly applicable as ours -- we can generate predictions for any given range of code, which has never been possible before.

Second, while most reviewers recognized the novelty of our approach, there were questions regarding the usefulness of predicting such (guard-like) invariants. This is a reasonable concern considering the breadth and novelty of our objective. We do emphasize that our model is demonstrably capable of detecting and repairing bugs. Some took issue with that evaluation being done in a fairly targeted fashion that assumed some bug localization; to those we note that such a setting is highly typical for evaluating current bug detection/repair work. We also argue, since some reviewers questioned the applicability of if-guards to invariant prediction, that the sheer overlap of our invariants with Daikon’s signals that our predicted statements are hardly biased towards if-guards alone.

Finally, we emphasize the timeliness of this line of work: an increasing number of papers in the last few years have focused on modeling program state as complementary to its syntax. We strongly concur that capturing state directly is of great benefit, and argue that our model’s ability to make general statements about that state can be instrumental in a wide range of downstream applications. As such, our approach may offer much potential in future explorations.

---

### Decision · Program_Chairs · 2021-01-07
**Final Decision**

**Decision:**

Reject

**Comment:**

The paper gives a way of constructing a dataset of programs aligned with invariants that the programs satisfy at runtime, and training a model to predict invariants for a given program.

While the overall idea behind the paper is reasonable, the execution (in particular, the experimental evaluation) is problematic. As a result, the paper cannot be accepted in its present form. Please see the reviews for more details.